# Discovery of a Nitric Oxide-Responsive Protein in *Arabidopsis thaliana*

**DOI:** 10.3390/molecules24152691

**Published:** 2019-07-24

**Authors:** Randa Zarban, Malvina Vogler, Aloysius Wong, Joerg Eppinger, Salim Al-Babili, Chris Gehring

**Affiliations:** 1Division of Biological and Environmental Science and Engineering, King Abdullah University of Science and Technology, 23955-6900 Thuwal, Saudi Arabia; 2Department of Chemistry, Technische Universität München, Lichtenbergstr. 4, 85748 Garching, Germany; 3College of Science and Technology, Wenzhou-Kean University, 88 Daxue Road, Ouhai, Wenzhou 325060, Zhejiang, China; 4Department of Chemistry, Biology & Biotechnology, University of Perugia, Borgo XX giugno, 74, 06121 Perugia, Italy

**Keywords:** *Arabidopsis thaliana*, nitric oxide, Heme Nitric Oxide/Oxygen (H-NOX) domain, Bric-a-Brac/Tramtrack/Broad Complex (BTB), NO-sensitive protein

## Abstract

In plants, much like in animals, nitric oxide (NO) has been established as an important gaseous signaling molecule. However, contrary to animal systems, NO-sensitive or NO-responsive proteins that bind NO in the form of a sensor or participating in redox reactions have remained elusive. Here, we applied a search term constructed based on conserved and functionally annotated amino acids at the centers of Heme Nitric Oxide/Oxygen (H-NOX) domains in annotated and experimentally-tested gas-binding proteins from lower and higher eukaryotes, in order to identify candidate NO-binding proteins in *Arabidopsis thaliana*. The selection of candidate NO-binding proteins identified from the motif search was supported by structural modeling. This approach identified AtLRB3 (At4g01160), a member of the Light Response Bric-a-Brac/Tramtrack/Broad Complex (BTB) family, as a candidate NO-binding protein. AtLRB3 was heterologously expressed and purified, and then tested for NO-response. Spectroscopic data confirmed that AtLRB3 contains a histidine-ligated heme cofactor and importantly, the addition of NO to AtLRB3 yielded absorption characteristics reminiscent of canonical H-NOX proteins. Furthermore, substitution of the heme iron-coordinating histidine at the H-NOX center with a leucine strongly impaired the NO-response. Our finding therefore established AtLRB3 as a NO-interacting protein and future characterizations will focus on resolving the nature of this response.

## 1. Introduction

Nitric oxide (NO) is a reactive oxygen species (ROS) that is utilized as a signaling molecule in many physiological processes in microorganisms, animals, and plants (for a review, see, e.g., [1]). NO in animals is well-characterized with its complex and pleiotropic roles in regulating immune responses to viral and bacterial infections [2,3], oncogenesis [4,5], and blood pressure maintenance [6], whereas, in plants, the roles and mechanisms of action of NO are currently less exhaustively described. Moreover, it has been proposed that NO signaling in animals and plants may share considerable functional similarity [7]. In plants, it has been demonstrated that NO acts as a critical regulator of development at all stages of the life cycle (for a review, see [8]). NO has also been implicated in plant responses to biotic [9] and abiotic stress [10,11]. Despite the many described NO effects in plants, our fundamental knowledge of NO production, sensing, and transduction in plants remains scarce. As a gaseous molecule, NO can diffuse freely across the membrane, so does not require canonical extracellular receptors for perception. Due to its ubiquitous nature, NO-sensitive or NO-responsive proteins, including, but not limited to, proteins that can bind NO in a reversible manner and proteins participating in redox reactions, are required to perform the aforementioned signaling functions. We have hypothesized that plants, much like bacteria and animals, contain proteins with heme-NO and oxygen-binding (H-NOX)-like domains [12] that can operate as NO-responsive proteins. In addition, we have recently demonstrated that an annotated *Arabidopsis thaliana* flavin monooxygenase (AtNOGC1; At1g62580) binds NO and has a higher affinity for NO than O_2_. Moreover, we have also shown that AtNOGC1 acts as a guanylate cyclase (GC) and that this activity can be modulated in an NO-dependent manner in vitro [13]. It has also been demonstrated that in an AtNOGC1 T-DNA insertion mutant, NO fails to induce stomatal closure, indicating that this “moonlighting” enzyme is central to both NO- and cGMP-dependent stomatal movements, thus establishing the biological relevance of direct NO-sensing [14,15]. It is important to note that AtNOGC1 harbors only the key residues of the H-NOX center that accommodate the heme group and not the full H-NOX domain. As such, NO-binding and NO-response of proteins containing H-NOX centers may be dissimilar from canonical H-NOX proteins found in other organisms.

Given the role of NO and the number of NO-dependent processes in plants e.g., growth and development, as well as defense responses, we hypothesize that plants contain NO-sensitive or NO-responsive proteins that can conceivably perceive NO. Here, we have applied a search motif constructed based on conserved amino acids at the heme b accommodating pockets of annotated NO-binding proteins across species and supported by structural modeling [16], in order to identify novel candidate NO-binding proteins in *Arabidopsis thaliana*. One of these proteins is a member of the Light Response Bric-a-Brac/Tramtrack/Broad Complex (BTB) family (LRB3; At4g01160) and was expressed and tested for responses to NO in vitro.

## 2. Results and Discussion

### 2.1. Identification of Candidate NO-Binding Proteins in Arabidopsis

Prokaryotes and eukaryotes harbor complex multi-domain proteins that accommodate functional centers in the form of catalytic centers or auxiliary sites required for binding to organic molecules and protein–protein interactions, which may serve as a highly specific and localized regulatory signal in complex proteins. As such, these seemingly hidden functional centers often fall below the detection limit of BLAST searches. However, such centers may be identified in proteins by applying search motifs (terms) that are built based on consensus amino acids in annotated and experimentally-confirmed proteins across species [17,18]. This approach has been successfully applied for the discovery of plant guanylate cyclases (GCs) and adenylate cyclases (ACs), which cannot be identified by querying, e.g., the Arabidopsis proteome with annotated GCs or ACs from animals, fungi, or bacteria [19,20,21,22,23,24,25,26]. This search method has also enabled the discovery of an abscisic acid (ABA) binding site in an Arabidopsis guard cell outward rectifying K^+^ channel (GORK) that has subsequently been shown to be directly modulated by ABA [27]. Given the potential of a rational amino acid motif-based search supported by a structural modeling approach [18], we have extended its application to search for NO-sensing proteins in *Arabidopsis thaliana*.

To this end, we extracted an H-NOX motif from an alignment of the centers of NO-binding H-NOX domains from NO-dependent soluble GC of different species, including bacteria, blue-green algae, insects, and mammals [12]. This motif “Hx(12)Px(14,16)YxSxR” includes amino acid residues critical for heme b-binding (Figure 1A), such as the histidine, tyrosine, and asparagine from the heme-binding pocket.

When this motif is used to query the Arabidopsis proteome, it returns four NO-binding candidate proteins: At1g62580, At4g01160, At5g19160, and At5g57690. At1g62580 encodes a flavin monooxygenase that can bind NO [13], as well as having GC activity reminiscent of animal NO-sensing soluble guanylate cyclases [30,31]. At4g01160.1 is a member of the Light Response Bric-a-Brac/Tramtrack/Broad Complex (AtLRB3), while At5g19160 encodes a member of the Trichome Birefringence-Like (TBL) family of proteins. Finally, At5g57690.1 is a pollen-specific diacylglycerol kinase 4 (DGK4), which has been recently shown to play key roles in lipid signaling pathways that cross-talk with NO during pollen tube development [8,32]. In H-NOX domains of bacteria such as *Thermoanaerobacter tengcongensis*, the proline (P) residue at position 14 of the H-NOX motif contributes to structural “flattening” of the distorted heme, resulting in an increased affinity for oxygen [33]. When we omit this proline from the search term “Hx(27,29)YxSxR”, 34 Arabidopsis candidate NO-binding proteins are retrieved, hinting at a potentially substantial number of heme-binding proteins with a capacity for NO-binding (Table 1). It is interesting to note that several of these proteins have previously been recognised as part of processes that are directly or indirectly responsive to NO and they may indeed link NO signals to responses to pathogens, particularly pathogenesis-induced proteolysis [34,35]. Furthermore, we have found two transcription factors (TFs) in the list that contain the modified H-NOX motif. Incidentally, NO-dependent induction of TF-encoding genes has been reported [36] and direct modulation of TFs by NO has also been proposed [37]. Finally, we have also noted two RNA-binding proteins (RBP) in the list, At2G44710.4 and the aconitase 1 (At4G35830) [38], and this may point to a direct role of NO in RBP-mediated RNA processing.

In the next step, we chose to further investigate the Light Response Bric-a-Brac/Tramtrack/Broad Complex protein AtLRB3 (Figure 1B). This protein was inferred to have ubiquitin transferase activity and forms part of the CULLIN3-RING ubiquitin ligase complex that mediates ubiquitination and the subsequent proteasomal degradation of target proteins [39]. The protein contains 505 amino acids (aa) and the H-NOX motif spans from 357 (H) to 391 (R) (Figure 1B).

This protein was subsequently modeled (Figure 1C) and the region corresponding to the H-NOX motif could conceivably form a pocket where the key residues for heme b-binding, namely H357, Y387, S389, and R391, could all reside and penetrate into a distinct cavity. We subsequently named this region the H-NOX center. This H-NOX center possesses favorable spatial and charge patterns for heme b-binding. Notably, the H357 residue can coordinate the central iron ion of the heme cofactor [40,41] (Figure 1C), much like the H-NOX protein of *T. tengcongensis* [33]. NO binding can presumably displace the proximal histidine–iron coordination [42], leading to a subsequent loss of the heme moiety [43]. Furthermore, the conserved YxSxR signature (Figure 1B) downstream of H357 may establish hydrogen bonds with the carboxylate residues of heme b [33,44].

### 2.2. UV-Visible Spectra of Recombinant AtLRB3 in Response to NO

To establish the role of AtLRB3 (At4g01160) as an NO-responsive protein, we determined the incorporation of heme into the apoprotein and evaluated the NO response of the holoprotein using UV/Vis spectroscopy. The At4g01160 gene was expressed and the gene product was purified, followed by reconstitution with hemin. The full heme protein showed a distinct Soret peak at 408 nm. This peak is concentration-dependent and resembles the range of other proteins with histidine-ligated ferric heme b [45,46] (Figure 2A).

Importantly, in Figure 2B, we showed that unfolded protein, which is the His-tagged recombinant AtLRB3 before reconstitution with hemin (Figure 2B—green line), has no Soret peak, while His-tagged recombinant AtLRB3 after reconstitution with hemin (see Figure 2A) yielded a distinct Soret peak at 408 nm. Spiking the buffer as well as heme-containing protein with free hemin yielded a Soret peak at 436 nm and a broad shoulder around 400 nm, which is distinct from the bound heme at 408 nm. The addition of hemin to the unfolded protein did not yield a Soret peak at 408 nm, but only peaks stemming from hemin in buffer alone (compared to hemin in buffer–black line in Figure 2B). In the same manner, BSA containing no His-tag and no heme-binding (inset of Figure 2B) did not exhibit a peak between 350 and 600 nm, and showed the same spectra as denatured AtLRB3 and with added hemin in the solution. As for unfolded protein, the addition of free hemin to BSA gave a peak corresponding to free hemin alone in buffer (Figure 2B). The unfolded protein and BSA controls further show that the peak at 408 nm is due to the true binding of heme to recombinant AtLRB3, resulting in a Soret peak similar to the nature of heme-containing proteins. It is important to note that the unfolded protein and BSA protein controls were performed in the same background as AtLRB3, with the exception that the unfolded protein contains a denaturing amount of urea (8 M) (see methods for full buffer composition). In addition to the use of unfolded protein and BSA as protein controls, the hemin-reconstituted AtLRB3 was also concentrated in 30 kDa Amicon Ultra centrifugal filter units (Sigma-Aldrich Corp., Missouri) and dialyzed against hemin-free buffer, using size exclusion to remove any carry-over free hemin from the purification step, thus eliminating the possibility of unspecific binding. In agreement with literature data for other H-NOX proteins [33,43], chemical reduction of the heme protein by sodium dithionite leads to a shift of the Soret peak to 418 nm and the appearance of β- and α-peaks at 525 and 553 nm (Figure 3A).

Next, we used UV/Vis spectroscopy to study the NO response of AtLRB3 under anaerobic conditions. Presumably, the native state of cytosolic AtLRB3 in *Arabidopsis thaliana* is ferrous, due to the reducing redox potential maintained by the glutathione pool [48]. Therefore, NO was added to the reduced AtLRB3. The addition of increasing amounts of NO in the form of diethylamine/nitric oxide complex (DEA NONOate) immediately after reduction by dithionite, led to a decrease in absorption at 418 nm accompanied by the emergence of a peak at 398 nm. Correspondingly, the α- and β-peaks of the reduced protein disappeared (Figure 3A). NO binding to H-NOX domains generates a transient 6-coordinate (6c) Fe(II) intermediate, with histidine as a proximal and NO as a distal ligand. According to recent studies, the strong covalent Fe(II)−NO σ-bond results in a marked thermodynamic σ-trans effect of NO, which greatly weakens the proximal Fe−NHis bond in six-coordinate ferrous heme-nitrosyls [49]. Furthermore, the His-bound heme is distorted from planarity due to steric interactions within the heme-binding pocket, which induces further strain of the His−Fe(II) bond. Accordingly, the histidine is displaced easily, resulting in a stable 5c-nitrosyl-heme complex [50]. This complex has been reported to absorb at 398 nm [51,52], as seen in the UV/Vis spectra here. With the displacement of the histidine ligand, the heme cofactor relaxes towards planarity and triggers a conformational change in the protein. This has been suggested as a key step in the mechanism of NO-induced signaling [53,54].

Generally, the observed spectroscopic behaviour measured for AtLRB3 resembles that of published H-NOX protein studies, such as the *S. oneidensis* H-NOX protein with Soret peaks at 403 nm (ferric), 430 nm (ferrous), and 399 nm (ferrous NO-bound form) [43,54,55]. Our measurements also correspond to reported values of 5c-Fe(II)porphyrin(NO) and 6c with an additional N-donor ligand (405 nm and 425 nm, respectively) [56]. It was previously suggested that particularly under high NO concentrations like in this study, a dinitrosyl complex could be formed after histidine displacement through the binding of a second NO. A labile dinitrosyl intermediate would quickly lose one NO [57]. Correspondingly, a crystal structure where a single NO is found to be bound to the proximal site, instead of the distal site, has been reported [54]. However, while either of the 5c heme-nitrosyl species (distal and proximal) should exhibit very similar UV/Vis spectroscopic characteristics, the transient 6c-intermediate should exhibit a Soret peak around 420 nm [54], which was not observed in our study.

We also monitored the behaviour of AtLRB3 after NO-saturation (post-NO saturation, pNS) and a reappearance of the Soret peak at 408 nm indicated the slow release of NO. Concomitantly, the broad absorption at 530 nm reappeared. The UV/Vis signature indicates oxidation of the Fe(II) center by NO and re-coordination of the proximal His to the ferric heme, being consistent with the release of NO from AtLRB3 (Figure 3B). Both the rebinding of the proximal histidine and a ferric NO-bound state have been suggested to play an important role in NO-release from cytochrome proteins. As the NO-heme interaction is thermodynamically weaker for the oxidized ferric state, it might play a role in the deactivation and dissociation of NO [58], e.g., for nitrophorins. Likewise, in NOS proteins, the ferric state can be involved in the release of NO [59,60]. However, the redox-potential of -0.76 V (NO to triplet nitroxyl anion, ^3^NO^-^) renders NO inert to reduction under physiological conditions and correspondingly, a heme-oxidation mechanism based on an Fe(II) to NO one-electron transfer is unlikely [61]. In fact, the strong σ donation of NO induces the transfer of electron density to the iron center of 5c heme nitrosyl complexes, resulting in a partial reduction of the metal and a noticeable Fe(I)-NO^+^ character [59].

In contrast, the reduction potential of the metastable (NO)_2_ dimer, which is present in relevant amounts at the higher NO concentration used in this study, is predicted to be quite favorable for heme oxidation (+0.33 V for (NO)_2_ + 2 e^−^ + 2 H^+^ → N_2_O + H_2_O). Indeed, the reduction of NO to N_2_O by cytochromes in the ferrous state was demonstrated to occur at NO concentrations above 1 mM, either through the 6c-bis(NO) state [62], an (NO)_2_ complex [55], or the reaction of a second free NO molecule with the reduced NO-heme complex [63,64]. Excess reducing agent, or any other external source, might provide the additional electron required by all these mechanisms.

As AtLRB3 was identified based on the conserved amino acids at the centers of H-NOX proteins across species, we probed the H-NOX center of AtLRB3 by site-directed mutagenesis, in order to check if there was indeed any reduced affinity for NO when one or more key amino acids were mutated. Therefore, we generated an AtLRB3/H357L mutant, where a leucine replaced the heme-binding histidine, and investigated its NO binding capability. We selected this mutation as histidine at the H-NOX center was determined in previous studies to be the proximal ligand for the iron of the heme cofactor to which NO binds as a distal ligand [49]. Leucine was chosen to replace histidine because it is more similar to histidine in terms of size compared to, e.g., alanine or glycine, but lacks the electrically-charged side chain of histidine which, we believe, allows histidine to ligate with heme better than leucine. Qualitatively, the UV/Vis spectra recorded under dithionate reduction with subsequent addition of the NO source are similar to those of the WT protein, presumably because the H-NOX center is structurally unchanged by this point mutation. The ferrous Soret peak at 418 nm disappears, while a new signal emerges at 398 nm, for the protein with coordinated NO. However, approximately a twofold higher concentration of the NO source is needed to obtain NO-saturation (Figure 4A), while disappearance of the signal at 398 nm occurs much faster than for the WT protein, with the Soret peak of the His-coordinated ferric heme at 408 nm forming after 2.5 h (Figure 4B) compared to longer than 6 h for the wildtype protein, thus assigning H357 a role in conferring NO affinity in AtLRB3. Since the NO ligand has a π-acceptor character, it binds more strongly to a heme coordinated by an electron density providing histidine [49]. We also checked for the possibility of S-nitrosylation using the S-NitrOsylation Site (SNOSite) on the cysteine prediction tool available at http://csb.cse.yzu.edu.tw/SNOSite/Prediction.html [65] and found two cysteine residues, C372 and C383, within the H-NOX center with a high probability of being S-nitrosylated. The specificity level for the prediction was set to “high (95%)”. However, our mutagenesis study involving the AtLRB3/H357L mutant clearly showed a diminished NO-binding response in identical experimental conditions as the WT protein ran parallelly during the assays, thus specifically assigning this NO response to the heme at the H-NOX center and alleviating concerns of S-nitrosylation interfering with the observed NO spectral changes.

We note that in canonical H-NOX proteins, mutating the histidine residue to alanine resulted in the complete loss of the Soret peak. We observed that while the H357L AtLRB3 mutant exhibits spectra identical to the WT, the NO-binding response was clearly diminished, and we have explained this observation by the π-acceptor character of NO. This dissimilarity may also be due to the fact that AtLRB3 only contains key residues in the H-NOX center and does not possess the entire H-NOX domain of canonical H-NOX proteins, which has naturally raised questions regarding its physiological relevance. There is indeed emerging evidence of the biological relevance of such H-NOX centers. For instance, a recent article reported that a protein from *Arabidopsis thaliana*, DGK4, harbors an identical H-NOX center, and was shown to be essential for normal pollen tube growth and for the overall development of the plant [32]. The authors also discussed the possibility of DGK4 playing signaling roles that cross-talk with NO and lipid pathways in the pollen tube. Furthermore, DGK4 has been previously identified by a Bioinformatics approach to contain an H-NOX center identical to AtLRB3 and was also speculated to play important NO signaling roles in the directional growth of the pollen tube [8]. Another Arabidopsis protein harboring such an H-NOX center, AtNOGC1, was previously shown to be NO-sensitive in vitro [13] and was later shown to be involved in NO-dependent and cGMP-mediated stomatal closure in Arabidopsis plants [14]. We are beginning to understand the physiological relevance of plant proteins harboring a seemingly reduced H-NOX domain and this discovery-centered report of AtLRB3 will pave the way for subsequent plant physiology works.

The collective experimental data suggest that the AtLRB3 protein contains a heme cofactor that is ligated by histidine 357. We could demonstrate the heme-protein’s ability to function as an NO-responsive protein in vitro, showing NO binding and the plausible formation of a five-coordinate intermediate after histidine displacement, which is required for the NO-dependent signaling of canonical H-NOX proteins. An H357L mutation in the heme-binding pocket results in a diminished NO affinity, and that further supports the assignment of AtLRB3 (At4g01160) as an NO-responsive protein most likely operating through an H-NOX-like pocket. While AtLRB3 harbors an H-NOX-like pocket, AtLRB3 is structurally dissimilar to canonical H-NOX proteins. Therefore, we can expect that the nature of this NO-response, when revealed, may be unique to complex multi-domain plant proteins. Therefore, it is also plausible that AtLRB3 belongs to a new class of heme-sensing proteins with flexible heme binding properties instead of being a canonical NO-sensitive protein. As such, our report can guide subsequent characterization studies on AtLRB3 and possibly also other plant proteins with similar H-NOX centers.

### 2.3. The Role of Light Response BTB Proteins

Bric-a-Brac/Tramtrack/Broad Complex (BTB) family proteins have been reported to direct the selective ubiquitylation of proteins after their assembly into Cullin3-based ubiquitin ligases (for a recent review, see [66]). The multi-subunit Cullin-RING Ligases (CRLs) are a highly polymorphic collection of E3s composed of a Cullin (CUL) backbone subunit onto which the E2-Ub-docking RING Box1 (RBX1) protein and a diverse assortment of adaptors that recruit ubiquitylation substrates are assembled [67]. One prominent CRL subtype incorporates the Bric-a-Brac/Tramtrack/Broad Complex (BTB) E3s. These contain RBX1, the CUL3 isoform, and a member from the large family of BTB target-recognition adaptors that docks with CUL3 via a signature approximately 95-amino acid BTB domain [68,69,70].

Recently, a subfamily of nucleus-localized BTB proteins that encode the LIGHT-RESPONSE BTB proteins (LRB) in Arabidopsis (*Arabidopsis thaliana*) has been described [71], and it was demonstrated that LRB1 (At2g46260) and LRB2 (At3g61600)) affect photomorphogenesis. It was also reported that *lrb1* and *Irb2* double mutants display a marked hypersensitivity to red light, but not to far-red or blue light, and are compromised in multiple photomorphogenic processes, and that red light hypersensitivity can be overcome by eliminating phytochrome B (phyB) and phyD from which an effect downstream of these two photoreceptor isoforms was deduced, possibly by affecting the turnover of phyB/D [71]. Our NO binding protein, which is annotated as LRB3, shares the domain organization with LRB1 and 2, and has a >50% sequence identity with 1 and 2, but does not appear to be directly involved in light responses. LRB1 and 2 also do not harbor the H-NOX motif, thus implying that the gas-sensing role of LRB3 is highly specific. Both LRB1 and 2 have been shown to have a role in the protein stability of phytochrome B and D [71] and this in turn has implications for photomorphogenesis, whereas the LRB3, while sharing the domain organizations LRB1 and LRB2, has undetermined functions.

The following question then arises: is there evidence for interaction between the BTB domain containing protein and NO? There is an indication coming from the Kelch-like ECH-associated protein 1 (Keap1) that contains four distinct domains, including the N-terminal BTB domain [72,73]. In the case of Keap1, the authors have combined molecular modeling with phylogenetic, chemical, and functional analyses to demonstrate that Keap1 directly recognizes an NO sensor cluster of basic amino acids that facilitate S-nitrosation of C151 [72] and concluded that Keap1 is a specialized sensor that quantifies stress by monitoring the intracellular concentrations of NO (and Zn^2+^ and alkenals) serving as danger messengers. It is therefore tempting to speculate that LRB3 is a BTB family member that has sub-functionalized into an NO-responsive protein and eventually test if it can link NO signaling to changes in cellular protein levels, which is in turn linked to the specific turnover mediated by the ubiquitin/26S proteasome system [74].

## 3. Materials and Methods

### 3.1. Preparation of the Recombinant AtLRB3 (At4g01160)

Full-length cDNA of the *At4g01160* gene was synthesized by Invitrogen GeneArt^®^Gene Synthesis (Darmstadt, Germany) and cloned into a pENTR221 vector. LR recombination was used to transfer the gene into the vector pDEST17, in order to express the protein in BL21-AI *E. coli* cells. The pelleted bacterial cells were lysed buffer A (6 M guanidinium chloride, 20 mM Na_2_H_2_PO_4_, and 500 mM NaCl; pH 7.8). The expressed protein contained an N-terminal His-Tag for affinity purification with Ni-NTA agarose. The lysate was loaded into the pre-filled Ni-NTA column and washed three times with buffer B (8 M urea, 20 mM Na_2_H_2_PO_4_, 20 mM imidazole, and 500 mM NaCl; pH 7.8) and binding protein was eluted with buffer C (250 mM Imidazole added to buffer B). Prior to refolding using FPLC, the denatured purified protein was concentrated and then taken up into buffer D (7.5 M urea, 20 mM Na_2_H_2_PO_4_, 500 mM NaCl, 100 mM sucrose, 100 mM non-detergent sulfobetaines (NDSB), 0.05% (w/v), polyethylene glycol (PEG), 4 mM reduced glutathione, 0.04 mM oxidized glutathione and a protease inhibitor cocktail (SIGMAFAST); pH 7.8). Gradual removal of urea was obtained by gradient dilution of buffer D with buffer E containing hemin (50 mg/200 mL hemin, 1 M urea, 20 mM Na_2_H2PO_4_, 500 mM NaCl, 500 mM sucrose, 100 mM NDSB, 0.05% (*w*/*v*) PEG, 4 mM reduced glutathione, 0.04 mM oxidized glutathione, and the protease inhibitor cocktail; pH 7.8). The refolded protein was then washed again with 10 column volumes of buffer E and eluted with Buffer F (buffer E without hemin and with 500 mM imidazole) (see flowchart in Figure 5). Protein concentrations were determined by Bradford assays and all buffers were sonicated, filtered, and stored at RT. The AtLRB3 H357L mutant was generated by a site-directed mutagenesis PCR-based method using 5′ GCCTATGTACTCAGACCCATCAA 3′ forward and 5′ GATGGGTCTGAGTACATAGGCT 3′ reverse as internal primers designed with a single base substitution to alter the gene sequence. The presence of the mutation was confirmed by sequencing. 5′ ATGGATCTCTCACTCTCCGGT 3′ forward and 5′ CTAAAGGCGGATGGAGAGATC 3’ reverse primers were used to flank the end of the target sequence. The procedure for site-directed mutagenesis has been detailed elsewhere [75].

### 3.2. Computational Assessment of the AtLRB3 Heme-Binding Pocket

The crystal structure of an H-NOX protein from *T. tengcongensis* (PDB entry: 3TF0) served as the template for a full-length homology model of AtLRB3, which was generated using Modeller (ver. 9.14) [28] according to methods described previously [16]. All structures and images were analysed and prepared with UCSF Chimera [29]. Structural analysis of the heme-binding pocket of AtLRB3 was performed by probing the key residues within the H-NOX domain and was compared with that of the *T. tengcongensis* H-NOX protein.

### 3.3. Determining NO Binding In Vitro

Changes in the heme-accommodating domain during the binding of NO were followed using UV/Vis absorption spectroscopy. Here, spectral characteristics are strongly indicative of the type of heme which is present and are influenced by the heme environment, thereby providing a valuable tool for observing changes in the surrounding heme [72]. In more detail, the porphyrin structure contains an aromatic system of π electrons consisting of molecular orbitals which are occupied and unoccupied (π and π*). UV/Vis spectra show characteristic Soret peaks which stem from π → π* electron transitions [76,77].

To show heme b incorporation, the spectra of air-exposed protein (89 μg/mL), as well as spectra with added hemin (final concentration 61 μM), were measured using a TECAN Infinite M1000 microplate reader. Spectra of unfolded protein (500 μg/mL) and BSA (1 mg/mL) with or without added hemin were measured as a control.

NO binding was monitored in a glove box (InerTec AG) under nitrogen atmosphere using a TECAN Infinite 200 microplate reader. All solutions were flushed for 20 min with argon to remove oxygen before transfer into the glove box. The heme b-containing protein (400 μg/mL) was reduced chemically by adding sodium dithionite to a final concentration of 8 mM. Then, the NO donor diethylamine/nitric oxide complex (DEA NONOate) was immediately added in increments to the ferrous heme-protein. The Soret peak was monitored and DEA NONOate addition was continued until the absorption frequency remained constant, which indicates NO saturation. Over 24 h, repeated recordings of UV/Vis spectra were used to observe the slow diffusion of NO from the heme center.

## 4. Conclusion

We have used aligned centers of H-NOX domains from bacteria, blue-green algae, and mammals to extract a consensus motif based on functionally-assigned and conserved amino acid residues at the heme-accommodating pockets, in order to search for candidate NO-binding molecules in *Arabidopsis thaliana*. Furthermore, the selection of candidate H-NOX center-containing proteins was supported by structural modeling. We then demonstrated that a recombinant Light Response Bric-a-Brac/Tramtrack/Broad Complex (BTB) family protein binds NO with spectral characteristics reminiscent of canonical H-NOX proteins. Our data shows that AtLRB3 incorporates a histidine-coordinated heme cofactor with a characteristic Soret peak at 408 nm for the oxidized protein that shifts to 418 nm after chemical reduction with sodium dithionite. AtLRB3 binds NO, resulting in a 5-coordinate complex after histidine dissociation, as indicated by the Soret peak shift to 398 nm, reported in the literature for NO-bound 5-coordinate H-NOX proteins. Importantly, we have also demonstrated the reversibility of NO-binding by measuring spectra post-NO saturation indicating NO dissociation, while a mutant with a replacement of the heme-binding histidine with leucine yielded strongly reduced affinity for NO and thus showed slower binding of NO with faster dissociation. Taken together, the NO-binding dynamics of AtLRB3 and the AtLRB3/H357 mutant have assigned AtLRB3 as an NO-responsive protein. Further work will ascertain the exact nature of this NO response (e.g., acting as a sensor or participating in redox reactions), as well as its associated biological roles. If indeed more NO-responsive plant proteins of a similar nature will be discovered, many more important cellular and physiological functions directly or indirectly linked to NO might also be discovered.

## Figures and Tables

**Figure 1 molecules-24-02691-f001:**
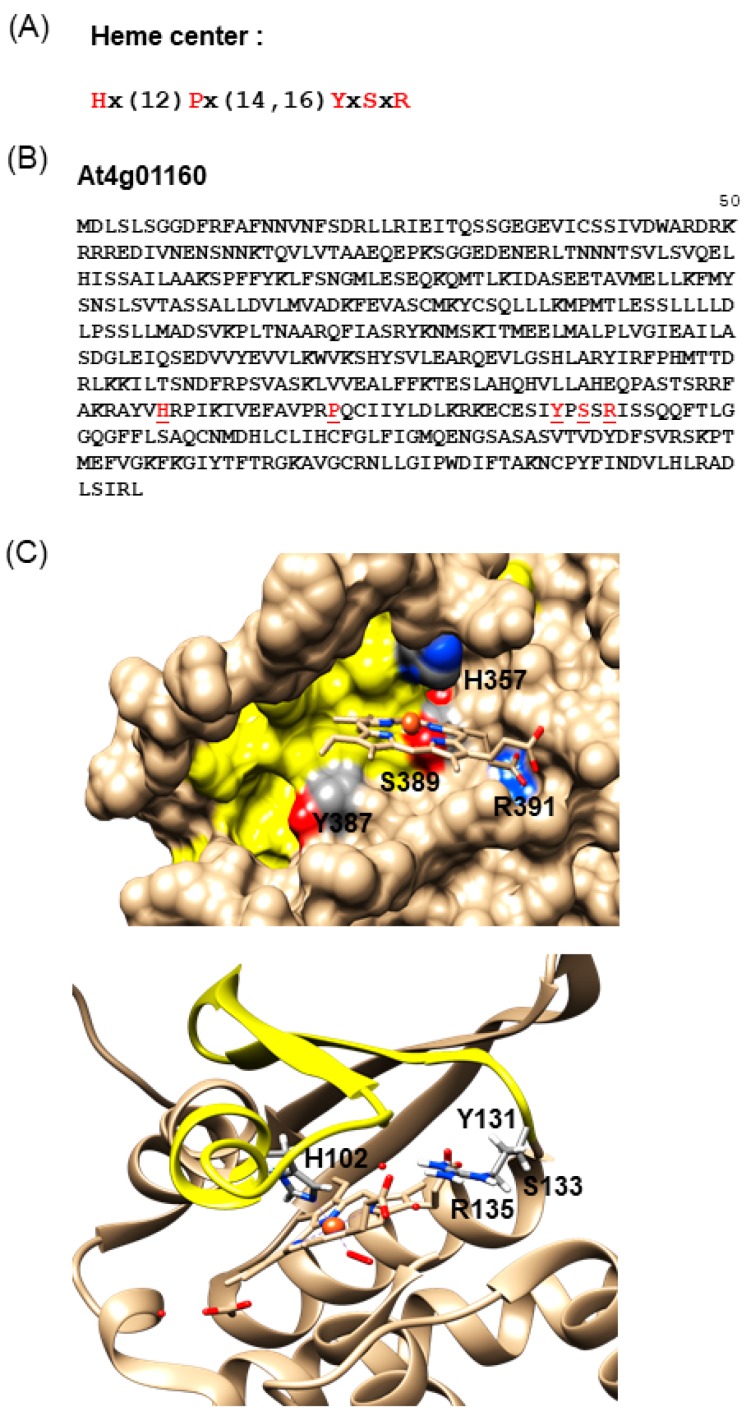
(**A**) Heme center search motif. (**B**) Amino acid sequence of Light Response Bric-a-Brac/Tramtrack/Broad Complex (AtLRB3), with the key residues in the heme center highlighted in red. (**C**) The homology model of AtLRB3 harbors a Heme Nitric Oxide/Oxygen (H-NOX) center (top) that can conceivably accommodate heme and is reminiscent of the H-NOX domain of *Thermoanaerobacter tengcongensis* (PDB entry: 3TF0) (bottom). The heme center is highlighted in yellow and key residues in the H-NOX domain are indicated in the 3D structures. The full length AtLRB3 model was generated based on the crystal structure of an H-NOX protein from *T. tengcongensis* (PDB entry: 3TF0). The AtLRB3 H-NOX heme center was assessed using Modeller (ver. 9.14) [28]. Molecular graphics and analyses were performed with the UCSF Chimera package [29]. Chimera is developed by the Resource for Biocomputing, Visualization, and Informatics at the University of California, San Francisco (supported by NIGMS P41-GM103311).

**Figure 2 molecules-24-02691-f002:**
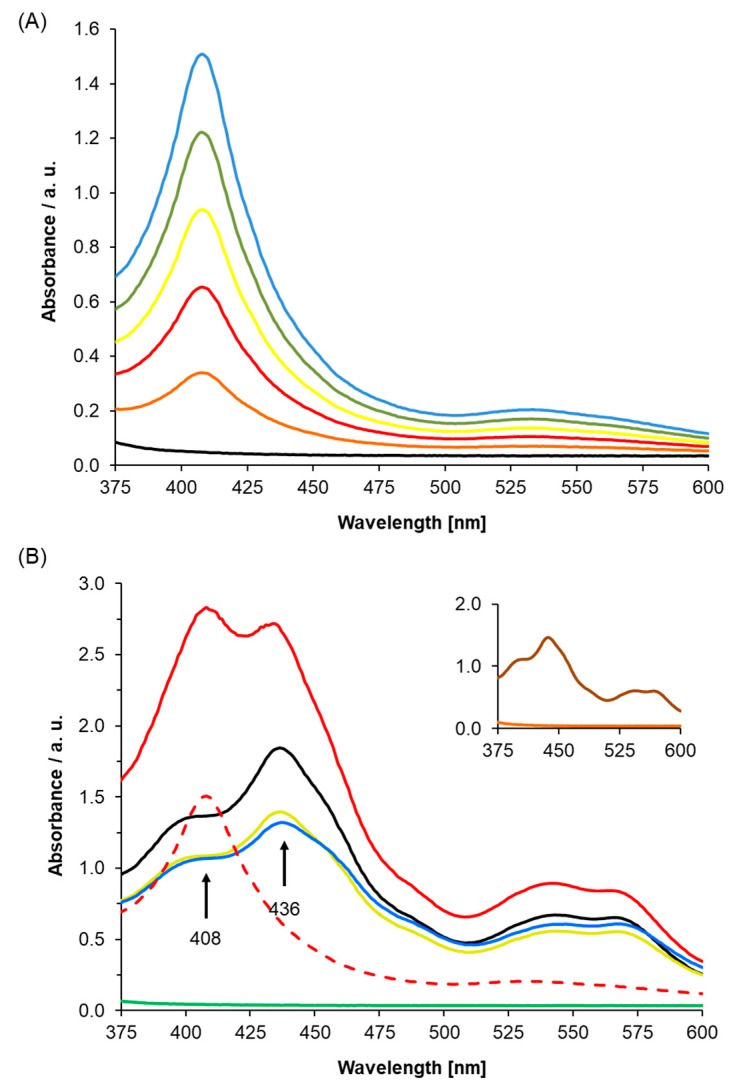
(**A**) UV/Vis spectrum of purified Light Response Bric-a-Brac/Tramtrack/Broad Complex (AtLRB3) under oxidizing conditions. The protein shows a distinct Soret peak at 408 nm, which is concentration-dependent (black: buffer F, blue–orange: AtLRB3 at 89/71/53/36/18 µg/mL). (**B**) Absorption spectra were measured after adding hemin to the purified folded protein (89 µg/mL) and unfolded protein (500 µg/mL) as a control (green: unfolded AtLRB3 without excess hemin, blue: unfolded AtRLB3 + hemin, red: heme reconstituted AtLRB3 + excess hemin, black: hemin in buffer F, light green: hemin in buffer C, dotted red: AtLRB3 without excess hemin). Hemin in both buffers (free hemin) showed Soret peaks at 436 nm, which were different from the Soret peak at 408 nm of hemin protein. Unfolded protein with added hemin shows an absorption peak similar to hemin in buffer alone (buffer compositions as described in the method). Inset: BSA (1 mg/mL) (orange) and BSA with added hemin (brown) as a negative control.

**Figure 3 molecules-24-02691-f003:**
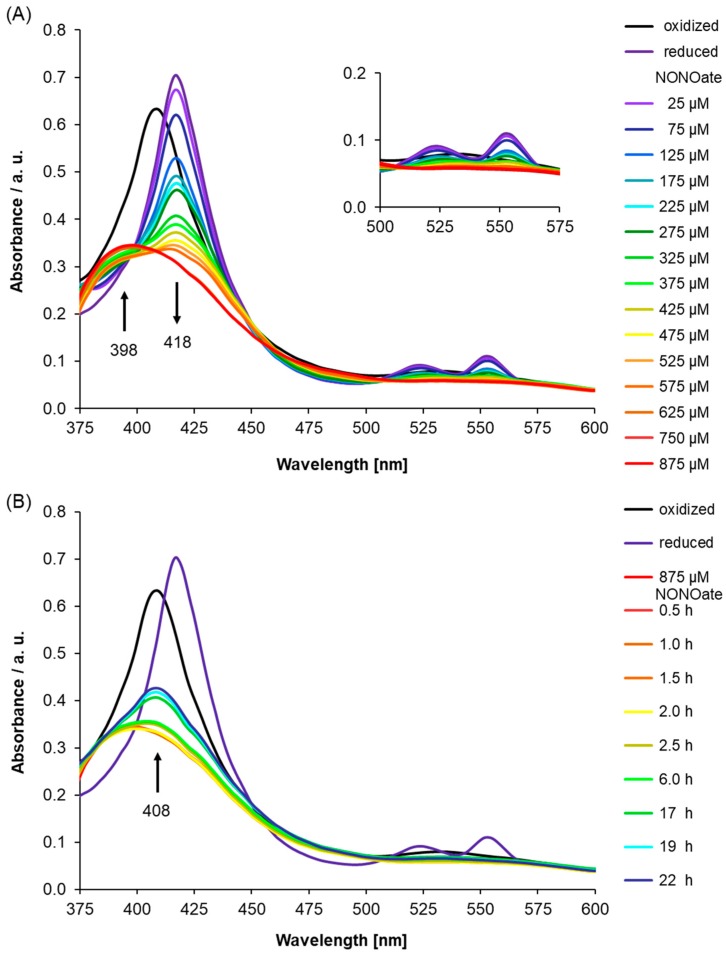
(**A**) Nitric oxide (NO) binding was monitored after the addition of diethylamine/nitric oxide complex (DEA NONOate) (NO donor) to reduced Light Response Bric-a-Brac/Tramtrack/Broad Complex (AtLRB3) (400 μg/mL protein) under anaerobic conditions. On reduction by sodium dithionite (8 mM), the Soret peak shifts by ~10 nm to 418 nm and the β- and α-peaks appear at 525 and 553 nm. Subsequent addition of NO results in a noticeable change of the UV/Vis spectra. The signature of the ferrous heme (Soret and β- and α-peaks) vanishes with an increasing concentration of the NO donor, while a broad peak at 398 nm appears, indicating a 5c nitrosyl heme center. Addition beyond 625 µM of NO donor does not result in further spectral changes, indicating the formation of NO-saturated protein. (black: oxidized, dark violet: reduced, violet–red: NO addition in increments) Inset: Highlights α- and β-peak regions. (**B**) The spectral behavior of AtLRB3 after NO-saturation (post-NO saturation, pNs) was monitored under anaerobic conditions. UV/Vis spectra were recorded at several time points. NO dissociates slowly from the heme and results in the reappearance of the Soret peak at 408 nm, which is characteristic of the oxidized (ferric) state of heme iron. UV/Vis spectra of the ferric and ferrous protein are included for comparison. (black: oxidized, dark violet: reduced, red: NO-saturated protein, dark red–blue: pNS measured at several time points). The inset of Figure 3A shows the characteristic α- and β-peaks of the oxidized and reduced protein. The observed Soret peak and split β- and α-peaks suggest the presence of an unknown sixth ligand coordinated to the ferrous heme center similar to the absorption values reported during the purification of sGC versions [47].

**Figure 4 molecules-24-02691-f004:**
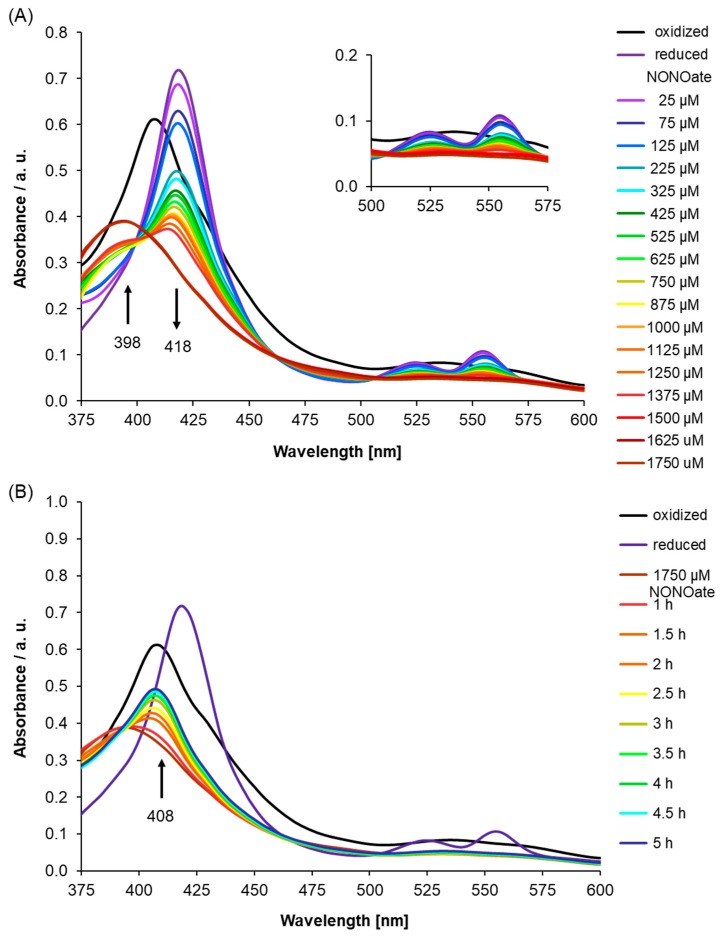
(**A**) Nitric oxide (NO) binding to an Light Response Bric-a-Brac/Tramtrack/Broad Complex (AtLRB3)/H357L mutant was monitored after the addition of diethylamine/nitric oxide complex (DEA NONOate) (NO donor) to reduced mutant AtLRB3 (400 μg/mL) under anaerobic conditions. After purification, air-oxidized protein shows a Soret peak at 408 nm, which shifts upon chemical reduction by sodium dithionate (8 mM) to 418 nm with the appearance of β- and α-peaks (525 and 553 nm). Addition of the NO donor leads to the formation of NO-bound protein showing a broad shoulder at 398 nm. NO affinity of the H357L mutant is reduced compared to the wild-type (see Figure 3). (black: oxidized, dark violet: reduced, violet–brown: NO addition in increments) Inset: Highlights α- and β-peak regions. (**B**) As for the wild-type protein, the behavior post-NO-saturation (pNs) was monitored for the AtLRB3/H357L mutant under anaerobic conditions. Absorption spectra were measured at several time points. Over time, this results in a reappearance of the Soret peak of oxidized heme iron at 408 nm. Compared to wild-type protein, diffusion occurs much faster. UV/Vis spectra of the ferric and ferrous protein are included for comparison (black: oxidized, dark violet: reduced, brown: NO-saturated protein, red–blue: pNs spectra measured at several time points).

**Figure 5 molecules-24-02691-f005:**
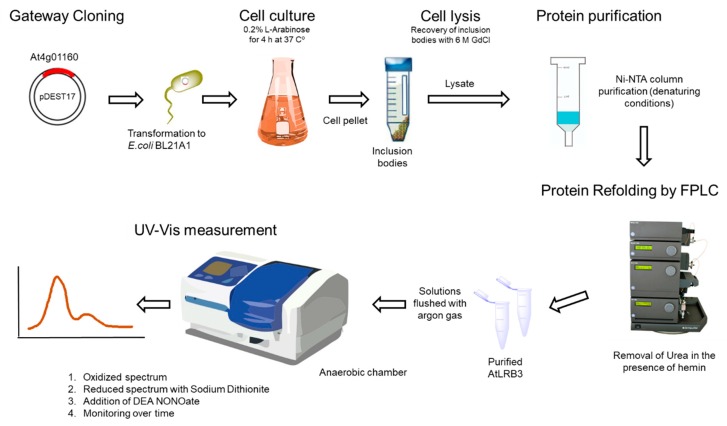
Flowchart for the generation of recombinant Light Response Bric-a-Brac/Tramtrack/Broad Complex (AtLRB3) proteins.

**Table 1 molecules-24-02691-t001:** *Arabidopsis thaliana* proteins that contain the following modified Heme Nitric Oxide/Oxygen (H-NOX) motif: Hx(27,29)YxSxR.

ATG number	Annotation	NO
At1G08630.7	Threonine aldolase 1	
At1G13940.1 ^TF^	T-box transcription factor, putative (DUF863)	
At1G55870.1	ABA-hypersensitive germination 2	**+** ^1^
At1G74800.1	Galactosyltransferase 5, GALT5	**+** ^2^
At2G28250.1	Protein kinase superfamily protein;	
At2G42280.1 ^TF^	ABA-resp. kinase substrate 3—(bHLH) DNA-binding fam. prot.	**+** ^3^
At2G44710.4 ^RBP^	RNA-binding protein (RRM/RBD/RNP motifs)	
At3G06260.1	Galactinol synthase 9	**+** ^4^
At3G29170.1	Transmembrane protein (DUF872)	
At3G57420.1	Regulates assembly & trafficking of cellulose and synthase complexes	
At4G01160.2	Encodes a member of LRB BTB family	**++**
At4G17250.3	Transmembrane protein	
At4G26330.2 *	Subtilisin-like serine endopeptidase family protein	**+** ^5^
At4G31570.3	Nucleoporin	
At4G35830.1 ^RBP^	Aconitase 1	**+** ^6^
At4G36195.1	Serine carboxypeptidase s28 family protein	
At4G39290.1	Galactose oxidase/KELCH repeat superfamily protein	
At5G02310.2 *	Proteolysis 6, PRT6 subtilase family protein	
At5G11940.1 *	Subtilase family protein	
At5G52850.1	Pentatricopeptide repeat (PPR) superfamily protein	
At5G53280.1	PVD1, plastid division1	
At5G67170.1 *	SEC-C motif-containing protein/OTU-like cysteine protease family	

**+ ** Work done in plants; **++ ** Nitric oxide (NO)-interaction confirmed in this work. ***** Proteins with a role in proteolysis; TF: transcription factor; RBP: RNA-binding protein. ^1^ Direct and indirect evidence of NO-dependence of abscisic acid (ABA)-mediated processes. ^2^ Evidence that NO production (e.g., during seed germination) affects GALT5 expression. ^3^ The nitric oxide promoted light-induced seed germination that requires a bHLH (PIF1). ^4^ Specifically pollen and hence NO modulated expression. ^5^ Subtilisin-like proteases have been identified as S-nitrosylation targets. ^6^ Aconitase 1 is inhibited by NO, thereby affecting the conversion of citrate to isocitrate and hence the TCA.

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
