# Peer review of "Discovery of a Nitric Oxide-Responsive Protein in *Arabidopsis thaliana"

_molecules, 2019, doi:10.3390/molecules24152691_

Round 1
Reviewer 1 Report
The manuscript is significantly improved, however, the NO dissociation still is scientifically unsound. You cannot claim to be measuring NO dissociation in these experiments, for at least 2 reasons. 1) You cannot measure NO dissociation from a heme protein without a NO trap - NO association rate constants Also, NO dimers are only favored at high pressure, which is not present in your experiments, as far as I can tell, so the explanation for iron oxidation is unlikely. I think the authors would be better off leaving this experiment out altogether unless then can go back and do it properly. It would not take any more protein to do a NO dissociation measurement with the CO/dithionite trap than the experiment they currently report.
I think the authors misunderstood my previous comments about the shoulder on the Soret band. Figure 2B shows multiple peaks upon heme addition, presumably due to free heme and protein bound species, but there appears to be at least 3 peaks in the Soret region of the spectrum - it is unclear what the authors are trying to demonstrate with this figure. Especially since a very similar spectrum is present with BSA + heme, which was meant to be a negative control (I think), so it seems that this figure contradicts the authors claim that their protein is a specific heme binding protein.
Exactly what is happening in figure 3 is also still unclear. With excess dithionite and DEA/NO in solution, why don't they see significant absorbance in the lower part of the spectrum - is this why the spectrum is cut off at 375 nm? Still, one would expect a shoulder from the dithionite peak at 375 nm. Also their is no clear isosbestic point, which would indicate you are not going from cleanly from Fe(II) to Fe(II)-NO - again, the Soret region appears to include multiple species. The spectra for the mutant H357L actually look cleaner than the wild-type, which doesn't really make sense if that heme is the heme ligand. Perhaps this protein is not an NO-sensitive protein, but in stead one of the new heme-sensing proteins being hypothesized with flexible heme binding properties?
Reviewer 2 Report
I was not one of the 2 people who initially appraised the manuscript but I read the paper carefully as well as the answers to the experts' comments and questions. Although the authors were not able to answer all the points requested in particular by reviewer 1, the manuscript has been substantially improved compared to its first version.
I have some minor comments:
- a word is missing from line 126
- I wanted to know what is the threshold used for SNOSite analysis (high, medium or low?)
Reviewer 3 Report
In the paper entitled “Discovery of a NO-sensitive protein in Arabidopsis thaliana”, the authors have first identified in silico potential candidates of H-NOX centre containing proteins. Then they demonstrated that a recombinant Light Response Bric-a-Brac/Tramtrack/Broad Complex (BTB) family protein, i.e. AtLRB3, binds NO with spectral characteristics similar to canonical H-NOX proteins. They show that AtLRB3 incorporate a histidine-coordinated heme cofactor and that binds NO resulting in a 5-coordinate complex after histidine dissociation. Finally, they show that replacement of the heme-binding histidine with leucine, is associated to a marked reduction in the affinity for NO.
The study is a very interesting preliminary report on a potential NO-responsive protein in Arabidopsis thaliana. However, the demonstration that AtLRB3 is actually an NO-sensor is missing. Further work should be focused to prove that NO-binding to AtLRB3 is able to trigger a signaling cascade with physiologically-relevant functions. The study is well-executed, and opens up for more in depth studies on the role of NO in plant physiology. However, the title seems, at this stage of the research, a little misleading and should better read as “Discovery of a potential NO-binding protein in Arabidopsis thaliana”.
This reviewer will consider this manuscript acceptable pending this minor issue.
Round 2
Reviewer 1 Report
As mentioned in my 2 previous reviews, I think the oxidation of heme iron upon NO dissociation under aerobic conditions is curious. I do not think it is likely that NO dimers are being formed under the experimental conditions, especially since excess reluctant is invoked in this mechanism to oxidize heme.
Author Response
"Not applicable"
This manuscript is a resubmission of an earlier submission. The following is a list of the peer review reports and author responses from that submission.
Round 1
Reviewer 1 Report
Major concerns:
1. More detail is needed in regards to how some of the experiments were performed. Without these details, the current experimental design appears to be flawed. However, it is possible that clarification could relieve these issues, and that crucial and necessary details were simply left out.
a. How were the various hemoprotein complexes (FeIIand FeII-NO) made? After addition of sodium dithionite to AtLRB3, was excess dithionite removed? After addition of the NO donor DEA NONOate, was excess DEA removed? The proper controls must be included to ensure that changes in spectra are not a result of excess DEA/ dithionite being present in the samples. In addition, an excess of dithionite in the sample will inhibit the formation of the NO bound complex. This would explain why such a high amount of NONOate was required to form the NO bound complex. Figures 3A and 4A, which demonstrate the concentration of DEA NONOate required to form the AtLBR3-NO complex are inaccurate because dithionite was present.
2. More suitable experiments are necessary to prove AtLRB3’s role in NO sensing. Just because it binds NO doesn't mean it is an NO sensor. This protein appears to reduce NO (or at least why does NO dissociate to leave the ferric protein)? The experiments provided do not provide sufficient information to determine NO association or dissociation kinetics.
a. AtLRB3 was purified under denaturing conditions, and hemin was added upon refolding. Why wasn’t the protein purified under native conditions? Most H-NOX proteins purify from E. colibound to heme. It would only strengthen the authors argument, that AtLRB3 is a hemoprotein, to show that AtLRB3 purifies heme bound.
b. To prove that AtLRB3 is a physiologically relevant NO sensor it is crucial to show NO binding is selective. Have the authors tested if AtLRB3 binds oxygen? What about carbon monoxide?
c. The experiments conducted to observe the dissociation of NO involved monitoring the formation of the ferric complex. This is unusual, as one would expect the presence of the ferrous complex upon NO dissociation. While the authors speculate that (NO)2 could be responsible for the oxidation of heme, trace amounts of oxygen in the glove box could do the same thing. Some NO binding hemoproteins that do not bind oxygen, will instead oxidize to their ferric form in the presence of oxygen. Or perhaps the protein reduces NO, leaving the ferric form? Why is the ferric form observed at all if there is excess dithionite? The methods are very difficult to understand and analyze.
d. Because of the issues mentioned above in regard to the NO binding experiments, a more suitable experiment such as a CO/dithionite trap should be used to determine NO dissociation rates (see reference 7). This would also eliminate the possibility that NO is re-associating with the heme to produce an artificially slow dissociation rate. A true rate of NO dissociation from AtLRB3 would be useful, in order to compare it to other known heme binding NO dissociation rates. In addition, it is necessary to provide statistical repeats for these types of experiments.
e. What is the association rate for NO binding?
3. How do the authors know that AtLRB3 protein is completely heme bound during the NO binding experiments? How much hemin needs to be added to ensure that all heme binding sites on are occupied? Is heme binding 1:1? If hemin was in fact determined to be added in excess, unbound hemin was removed by dialysis and size-exclusion chromatography, but what was final concentration of heme bound to protein.
Minor comments:
Line 355: “To show heme b incorporation, spectra of air-exposed protein (89 μg/mL), which is considered as fully oxidized (ferric)…” Reference? Be careful with the wording of this sentence. Aerobically purified H-NOX proteins are often a mix of ferrous and ferric forms and require treatment with potassium ferricyanide to fully oxidize.
Figure 2A: Data is shown from 375 to 600 nm. Were data points taken beyond 375 nm? Show the region including 280 nm to demonstrate total protein levels.
Figure 2B: Specify in figure caption that the red line is heme reconstituted AtLRB3 + hemin and not just folded apo AtLRB3 + hemin.
Figure 2B: How much hemin was added to each sample? The Soret maxima are all at different absorbance values. How was this amount of hemin determined?
Line 146: What is responsible for this broad shoulder at 400 nm? It should also be stated somewhere what the Soret, α, and β peaks represent.
For the NO binding experiments, what is the final concentration of hemin that was added to the protein? How was this amount determined to be suitable?
H357 Is one of many histidine’s (11) in AtLRB3, and considering that the mutation of H357 to leucine did not impact the Soret, have the authors looked at other histidine residues? It would be useful to provide a sequence alignment of AtLRB3 against previously studied H-NOX proteins, for which the heme ligating histidine residue has been identified, to show which residues are conserved.
Reviewer 2 Report
In this manuscript authors identify candidates for NO-binding proteins in Arabidopsis thaliana by a search term constructed based on conserved and functionally annotated 20 amino acids at the centres of Heme Nitric Oxide/Oxygen (H-NOX) and briefly characterize the protein AtLRB3 (At4g01160).
The findings of this manuscript are potentially very interesting as NO-sensitive/responsive proteins in plants still needs further work. Some concerns and queries have to be addressed, however:
1. May authors add a figure with the purification process of the protein? How purified it is?
2. May authors add the table of the putative 61 candidates for NO-binding when omitting Proline in the search?
3. As it appears that there are some Cys around the pocket, it is possible that the protein may be also target of S-nitrosylation, as haemoglobins do?. If the authors have the purified protein is not difficult to make a biotin-switch for example.